# Vaccinations Status against Vaccine-Preventable Diseases and Willingness to Be Vaccinated in an Italian Sample of Frail Subjects

**DOI:** 10.3390/vaccines10081311

**Published:** 2022-08-14

**Authors:** Caterina De Sarro, Rosa Papadopoli, Maria Carmela Morgante, Carmelo Giuseppe Angelo Nobile, Giovambattista De Sarro, Claudia Pileggi

**Affiliations:** 1Department of Health Sciences, University “Magna Graecia” of Catanzaro, 88100 Catanzaro, Italy; 2Department of Pharmacy, Health and Nutritional Sciences, University of Calabria, Arcavacata of Rende, 87036 Cosenza, Italy; 3FAS@UMG Research Center, Department of Health Science, School of Medicine, University of Catanzaro, 88100 Catanzaro, Italy

**Keywords:** frail patients, recommended vaccination, missed opportunities

## Abstract

Background: Study aim was to investigate the vaccination status against vaccine-preventable diseases (VPD) of frail adults during the SARS-CoV-2 pandemic and, for those subjects eligible for at least one vaccine, with respect to the recommended vaccination in line with the Italian National Vaccination Prevention Plane (NPVP), to explore the willingness to be vaccinated. Methods: A cross-sectional study was carried out among adults aged ≥ 60, immunocompromised or subjects affected by chronic conditions. Results: Among the 427 participants, a vaccination coverage rate lower than the targets for all the vaccines considered was found. Of those, 72.6% of subjects stated their willingness to receive recommended vaccinations, and 75.2% of the respondents stated that the advice to undergo vaccinations was received by the General Practitioner (GP). In a multivariable logistic regression model, higher odds of recommended VPD vaccination uptake (defined as having two or more of the recommended vaccinations) were associated with the willingness towards recommended VPD vaccination (Odds Ratio = 3.55, 95% Confidence Interval: 1.39 to 9.07), university education (OR = 2.03, 95% CI: 1.03 to 3.97), but having another person in the household (OR = 0.52, 95% CI: 0.28 to 0.97), and history of oncological disease (OR = 0.39, 95% CI: 0.18 to 0.87) were predictive of lower odds of vaccination uptake. In another multivariable model, higher odds of willingness to receive vaccines were associated with kidney disease (OR = 3.3, 95% CI: 1.01 to 10.5), perceived risk of VPD (OR = 1.9, 95% CI: 1.02 to 3.3), previous influenza vaccination (OR = 3.4, 95% CI: 1.8 to 6.5), and previous pneumococcal vaccination (OR = 3.1, 95% CI: 1.3 to 7.7), but increasing age (OR = 0.93 per year, 95% CI: 0.91 to 0.97), working (OR = 0.40, 95% CI: 0.20 to 0.78), and fear of vaccine side effects (OR = 0.38, 95% CI: 0.21 to 0.68) were predictive of lower odds of willingness to receive vaccines. Conclusions: Despite specific recommendations, vaccination coverage rates are far below international targets for frail subjects. Reducing missed opportunities for vaccination could be a useful strategy to increase vaccination coverage in frail patients during the routine checks performed by GPs and specialists.

## 1. Introduction

Prevention of infections is undoubtedly critical to ensure healthy ageing for the individual, and to reduce the socio-economic burden for societies. Vaccination is the main public health measure available for primary prevention of disease and it is the best defence against serious, preventable, and sometimes deadly, contagious diseases. Older age, geriatric syndromes, underlying chronic conditions, and multimorbidity have been recognised as major risk factors for severe outcomes in adults with vaccine-preventable diseases (VPD). Accordingly, the Italian National Vaccination Prevention Plan 2017–2019 (NPVP) [1] identifies specific categories, such as subjects with chronic conditions and the elderly population, that can benefit from free vaccinations.

Throughout 2011–2020, the Global Vaccine Action Plan (GVAP) [2] called on all countries to reach the 90% coverage threshold with all vaccines in the country’s national immunisation program by 2020. Although progress in vaccination coverage had already stalled before the pandemic began, the GVAP has provided a new global vision and a new strategy to influence national actions to implement immunisation at all ages. This approach has been taken up and repurposed by the Immunisation Agenda 2030 (IA2030) [3], which sets out very ambitious aims and, considering the lesson of COVID-19, puts much more emphasis on an immunisation strategy tailored to the national context and integrated into primary healthcare services.

This approach encompasses special groups at increased risk of VPD, such as patients with chronic and immune-compromising medical conditions, subjects taking immunosoppressive agents or immunomodulating, or those at increased risk of disease due to immunosenescence. In addition, during the SARS-CoV-2 pandemic, as soon as a vaccine against SARS-CoV-2 was available, the elderly population and frail subjects, in addition to healthcare workers (HCWs), were vaccinated as a priority due to the particularly high-risk of SARS-CoV-2 infection and subsequent severe complications. Recent studies performed on special cohorts of frail subjects, such as patients with Myastenia Gravis [4] or transplant patients treated with immunosuppressants [5], supported both the effectiveness of COVID-19 vaccines in the prevention of the worst complications (e.g., lethality, admission to Intensive Care Units) and the safety of these vaccines, which did not lead to serious adverse events (AE). In addition, previous studies have reported that influenza and pneumococcal vaccinations may be associated with a lower risk of COVID-19 mortality and hospitalisation [6,7,8]. Moreover, it has been estimated that 10% of COVID-19 deaths have been due to pneumococcal super-infection, and thus they could have been prevented by prior vaccination [9].

Therefore, in the light of these considerations, the aim of our study was to investigate the vaccination status against VPD of frail adults during the SARS-CoV-2 pandemic and, for those subjects eligible for at least one vaccine, with respect to the recommended vaccination in line with the NPVP among those indicated for their clinical conditions, to explore the willingness to be vaccinated.

## 2. Materials and Methods

### 2.1. Study Design and Population

The survey was conducted from February to May 2021, about two months after the start of the COVID-19 vaccination campaign. The Italian National Strategic Plan (INSP) [10] included the following categories as priority to receive COVID-19 vaccination due to the particularly high-risk of SARS-CoV-2 exposure, transmission, and clinical consequences: physicians and HCWs, long-term care patients and staff, elderly population, and frail subjects. The study population, consisting of older adults aged ≥60, patients affected by immune disease, immunocompromised or subjects affected by chronic conditions, was recruited among those attending a COVID-19 vaccination clinic waiting room. Immunocompromised subjects included those with primary or immunodeficiency disorders due to cancer, transplantation, asplenia, and immunosuppressive or immunomodulating medications. Subjects with chronic conditions included patients with cardiovascular, respiratory, neurological, renal conditions, diabetes, and chronic liver disease.

### 2.2. Survey Instrument

An anonymous questionnaire was administered to eligible subjects who consented to participate in the study. The questionnaire included four sections: in the first, sociodemographic information (sex, age, marital status, number in the household, education level, and employment status) were collected; in the second section, lifestyle habits and clinical conditions were investigated. To measure the burden of comorbid diseases, the Charlson Index was calculated for each patient. Furthermore, each patient was asked to indicate, on a scale ranging from 1 (very bad) to 10 (very good), the appropriate value representing their perceived health status at the time of the survey. In the third section, ten items explored the vaccination status and the eligibility with respect to the recommended vaccination in line with the NPVP and the recommendations of the Italian Society for Infectious and Tropical Diseases for adult vaccinations [1] and, specifically: Influenza; Pneumococcal; Meningococcal—Men ACWY; acellular Pertussis, in the combined formulation vaccine with Tetanus and Diphtheria—TDaP; Hepatitis B—HBV; Haemophilus influenzae type b—Hib; Herpes Zoster—HZV; and Measles, Mumps, Rubella, and Varicella—MMRV. Data on information received about recommended vaccinations and willingness to receive these vaccinations were also collected. In the last section, four items aimed to investigate the COVID-19 vaccination status (reasons related to the willingness to receive the immunisation, who recommended the vaccination, and, on a 10-point Likert scale, concern about the risk of contracting SARS-CoV-2 infection and fear of vaccine side effects). Data on vaccination status were referred to by patients.

The study protocol was ratified by the Regional Ethics Committee (ID No. 183).

### 2.3. Statistical Analysis

Statistical analysis was performed using the STATA software program (2016, Stata Corp, LP, College Station, TX, USA). Data were summarised by frequencies and percentages for categorical data and means and standard deviations (SD) for continuous data. The primary outcome of interest was the willingness to be vaccinated in subjects eligible for at least one vaccine among those indicated in the NPVP. The univariate analysis was performed using an appropriate test (*t*-test, χ^2^ test) to examine potential associations between the general characteristics and the willingness towards vaccinations.

Univariate and multivariate logistic regression analyses were performed to determine the independent association of explanatory variables with the following outcomes of interest: recommended vaccinations uptake in frail subjects, measured as having received at least two vaccinations among those indicated for their clinical conditions (Model 1) (0 = unvaccinated, 1 = vaccinated), and willingness towards recommended vaccinations (Model 2) (0 = unwillingness towards recommended vaccinations, 1 = willingness towards recommended vaccinations). The univariate analysis was performed using the *t*-test for continuous variables and χ^2^ test for categorical variables. 

The model-building procedure proposed by Hosmer and Lemeshow [11] was applied. The model only included variables with a *p* value of ≤ 0.25 in univariate analysis. Therefore, the following independent variables were included in both models: sex, age, total chronic health conditions, and visits to the GP in the previous year. Moreover, in Model 1 we also included: willingness towards recommended vaccinations, marital status, additional person in the household, educational level, neurological disease, oncological disease, respiratory disease, and how it was advised to carry out vaccinations against VPD; whereas in Model 2, the independent variables: working activity, smoking status, cardiovascular disease, dysmetabolic disease, kidney disease, the perceived risk of contracting VPD, having received vaccination information, fear of vaccine side effects, having received influenza vaccine in the previous three years, and pneumococcal vaccine in the past five years, were also included.

An adjusted odds ratio (OR) and 95% confidence intervals (CIs) were calculated.

## 3. Results

Patients eligible for the study were 451 and, of these, 24 refused to participate in the survey, for a response rate of 94.7%. No significant differences were found between the subjects who agreed to answer the questionnaire and those who refused, in terms of socio-demographic characteristics (sex, age, and number in the household).

The main characteristics of the study population regarding socio-demographic profile, health, and vaccination status are reported in Table 1.

The average age of the patients was 67.6 years (range: 20–95), with more than half (60%) being females, and 77.1% declared to be married. Overall, about one third were referred to as suffering from three or more chronic illnesses, and the most prevalent chronic health conditions were cardiovascular (74.9%), dysmetabolic diseases (40.1%), and oncological diseases (28.1%). Moreover, when patients were asked to indicate on a 10-point Likert-type scale their perceived health status ranging from 1 (very bad) to 10 (very good), the mean score was 6.9 ± 2.

Overall, 26.2% of the subjects declared they had not received information on recommended vaccinations according to their age and health status, while among the remaining 315 subjects who had received vaccine advice, 8.6% said they did not undergo any vaccination. For all the considered vaccines, the coverage was lower than the targets identified by the latest PNPV [1] as reported in Table 2.

Furthermore, most respondents (75.2%) stated that the advice to undergo vaccinations, included in the NPVP, was mainly received by the GP. Overall, more than two thirds of the patients (72.6%) stated their willingness to receive recommended vaccinations. Among the remaining 27.4% of subjects who declared they would refuse to receive recommended vaccinations, the most common reasons were the belief that vaccinations are dangerous (47.2%), lack of confidence in vaccines (16.7%), previous adverse events experienced after vaccine administration (11%) and having been poorly informed about the vaccines (8.3%). Moreover, when the perception of contracting VPDs was explored, more than one-third (37.6%) of respondents reported they did not believe they were at risk. All subjects enrolled in the study have received at least one dose of COVID-19 vaccination, and more than half (58.4%) of them reported that the recommendation to undergo the COVID-19 vaccination was mainly given by the specialists, whereas just a small percentage of them (16.2%) reported an indication by the GP. When reasons leading to the decision to undergo the COVID-19 vaccination were investigated, 74.3% of the patients declared the main driver was fear of getting the SARS-CoV-2 infection. Only 29.4% of subjects had low concerns about contracting the SARS-CoV-2 infection, and more than half (56.7%) declared that they were not at all worried about COVID-19 vaccine side effects.

Table 1 also shows the results of the univariate analyses, testing associations between characteristics of the study subjects (as hypothesised predictors) and self-reported vaccine uptake and stated willingness to be vaccinated (as outcomes).

In the final multivariable logistic regression model (Table 3), higher odds of VPD vaccination uptake (defined as having two or more of the recommended vaccinations) were associated with willingness towards recommended VPD vaccination (Odds Ratio = 3.55, 95% Confidence Interval: 1.39 to 9.07), and university education (OR = 2.03, 95% CI: 1.03 to 3.97), but having another person in the household (OR = 0.52, 95% IC: 0.28 to 0.97), and history of oncological disease (OR = 0.39, 95% CI: 0.18 to 0.87) were predictive of lower odds of vaccination uptake. In another multivariable model (Table 4), higher odds of willingness to receive vaccines were associated with kidney disease (OR = 3.3, 95% CI: 1.01 to 10.5), perceived risk of VPD (OR = 1.9, 95% CI: 1.02 to 3.3), previous influenza vaccination (OR = 3.4, 95% CI: 1.8 to 6.5), and previous pneumococcal vaccination (OR = 3.1, 95% CI: 1.3 to 7.7), but increasing age (OR = 0.93 per year, 95% CI: 0.91 to 0.97), working (OR = 0.40, 95% CI: 0.20 to 0.78), and fear of vaccine side effects (OR = 0.38, 95% CI: 0.21 to 0.68) were predictive of lower odds of willingness to receive vaccines.

## 4. Discussion

Although the SARS-CoV-2 pandemic has demonstrated the threat posed by infectious diseases and the relevance of vaccination as an extraordinary powerful tool to reduce the severity and mortality related to infections, only a few studies have addressed in depth the willingness to receive recommended vaccinations among elderly and frail subjects undergoing COVID-19 vaccination. This study has tried to explore the key factors associated with the achievement of one of the strategic objectives set by the GVAP [2], that every eligible individual should be immunised with all appropriate vaccines. 

The main research question investigated to what extent subjects have received recommended vaccinations in line with the NPVP and results revealed that target coverage rates are not being reached. As expected, influenza vaccination was reported by 70.5% of the patients, whilst only 21.1% had received pneumococcal vaccination in the five previous years. Several studies have been conducted to investigate the coverage rate among immunocompromised subjects; Pierron et al. [12], in a similar study conducted among haematological patients undergoing chemotherapy, reported higher rates for pneumococcal vaccination (45%) and lower for influenza (52%). A possible explanation of the higher rate of pneumococcal vaccination, as recognized by the authors, was an overestimation due to the conception of a single question about updating pneumococcal and dTPa vaccines, which made it impossible to specify which vaccine was up to date. In respect to influenza vaccination, the lower coverage shown by Pierron et al. [12] could be related to the physician’s adherence to the European guidelines [13], which did not recommend influenza vaccination in patients affected by haematological malignancies undergoing chemotherapy. In another French study [14], performed on frail subjects admitted to the emergency department, influenza vaccine coverage was lower than that observed in our population, with just 48.2% of the included patients, who reported influenza vaccination at least once in their lives. The authors proposed that the high percentage of missed opportunities for influenza vaccination by physicians and the complexity of the procedures leading to vaccination are possible explanations [14]. Very low influenza vaccination coverage rates have been reported in other European countries, such as Ireland (29.1%) [15] and Germany (23%) [16]. In line with our results, a study performed in the United States among patients with atherosclerotic cardiovascular disease reported that 32.7% of the sample had declared to have not received influenza vaccination in the 12 months prior to survey completion [17]. However, it seems useful to notice that we have collected information on at least one influenza vaccination that occurred in the previous three years, thus including the influenza campaign immediately preceding the availability of COVID-19 vaccination. It is plausible to hypothesize that the SARS-CoV-2 pandemic has been a major promoting factor with respect to previous campaigns and then could be considered the most important determinant of the increased adherence to influenza vaccination [18,19,20].

Although influenza and pneumococcal vaccination coverage was suboptimal in frail patients, when we assessed the willingness to get vaccinated versus VPDs, we found that it was significantly higher in participants who had already been vaccinated versus influenza and pneumococcal disease compared to unvaccinated, thus indicating that being vaccinated regularly could increase the adherence rate of vaccinations.

Most critical is the scene depicted when the other vaccinations, rather than influenza and pneumococcus, are considered. As regards to the HBV vaccine, our findings showed a very low coverage rate (8.3%), but in line with another Italian study that found a 9.2% vaccination rate among patients with liver cirrhosis [21]. Suboptimal self-reported HBV coverage was also revealed for ≥60 years old patients with diabetes, and the most relevant aspect was that no significant differences were highlighted in respect to subjects without diabetes [22], thus indicating the lack of attention of specialists in the routine assessment of their diabetic patients’ vaccination needs. 

Analogously, patients with high healthcare utilisation (e.g., number/year of inpatient admissions, emergency visits, outpatient visits) suffer delays in receiving the HZV vaccination [23], thereby determining a missed opportunity for vaccination (MOV) [24], resulting in a high healthcare burden at both the individual and societal levels [25,26]. Indeed, the elderly population and frail subjects are at a very high risk of neuronal damage due to the virus reactivation, which is responsible for the HZV pain. Neuropathic pain is frequently unresponsive to analgesic medications, persisting for several days, months, or even years after the onset of an HZV rash, causing considerable suffering and the need for healthcare services [27]. Therefore, the extremely low level of HZV vaccination coverage, highlighted in our previous research [28,29,30], underlines the urgent need for a policy for the improvement of vaccination coverage among frail subjects. Analogous considerations pertain to the other investigated vaccinations in frail subjects, such as Hib, which was reported by only 1.6% of the sample; Meningococcal vaccines (2.5%); TDaP (3.3%); and MMVR (6.8%).

Although alarming, these findings are not surprising; recent analyses of the burden of VPDs both in the United States [31] and in the European Union [32] suggest very high costs associated with control of these diseases, and these estimates are likely to escalate with the increase in the age and frailty status of the populations [33].

Moreover, when we measured the association between vaccine coverage and several variables, the educational level was associated with vaccine uptake. This finding is in line with a similar survey conducted among adults with chronic liver disease [34] in which age, educational level, and receipt of influenza vaccination were identified as factors associated with HBV vaccination.

Recently, Boey et al. [35] in a survey conducted to investigate the determinants of vaccination coverage in patients with underlying clinical conditions speculated that a possible reason for low coverage rates is that frail patients are closely monitored by specialists who often do not advise patients on vaccinations as it is considered a task of the GP; therefore, a vaccination recommendation from a specialist might have a considerable influence on the vaccination rate. Indeed, we found that more than half of the participants (58.4%) reported that the advice to undergo COVID-19 vaccination was provided by the specialist. Many concerns have arisen during the pandemic in frail patients affected by immunological conditions regarding vaccination (i.e., transplantation, myasthenia gravis, multiple sclerosis, oncological diseases) and the lack of data on this topic contributed to fueling the fear of vaccination, especially from GPs who were not aware of autoimmune conditions and their management. The case of myasthenia gravis is emblematic: in this condition, the lack of vaccination indirectly killed many patients who decided not to receive vaccination and faced COVID-19 without any protection, with consequent ICU admission and respiratory failure. Conversely, vaccinated myasthenic patients with subsequent COVID-19 presented lower rates of ICU admission and no deaths [4]. However, the COVID-19 vaccination campaign can be considered as a particular case, albeit emblematic of the role of the specialist in the meaningful implementation of vaccination compliance in frail patients, since in the initial stages, access to COVID-19 vaccination was allowed only to a few groups, including subjects aged 80 years or more and frail patients referred by specialists [10]. In contrast to what occurred for the COVID-19 vaccination, and in line with previous studies [36,37], the results revealed that advice to carry out vaccinations versus VPDs was provided by GPs. It has been frequently found [38,39] that patients identify GPs as the most reliable source of information about vaccination, and the most common reason for not being vaccinated is the lack of GP recommendation. Nevertheless, physicians often miss opportunities to recommend vaccinations to their patients [40,41] due to concerns about vaccination risks and a lack of trust in health authorities [40].

Getting vaccinated is the result of a complex series of behaviours, all of which are contingent on an interlocking system of thoughts and beliefs, people, funding, policies, and permissions [42]. Therefore, a comprehensive appreciation of both the possible impact of physical and cognitive frailty of the patient on non-vaccination-adherence is necessary for clinicians to improve the quality of care, in particular during the SARS-CoV-2 pandemic that continues to impact negatively on people’s quality of life [43,44], making it necessary to promote different adaptive interventions capable of making subjects more resilient in the current challenging times [45].

### Limitations

The limitations of this study must be considered. Patients were recruited among those attending the COVID-19 vaccination center; therefore, selection of patients with a higher positive attitude towards vaccinations may have occurred, thus overestimating the willingness to adhere to recommended vaccinations for frail patients. Another limit was that the vaccine coverage was calculated only with patients’ self-reported data, and as previously highlighted [46], there is a high risk of overestimating the vaccine uptake in respect to the use of medical records. However, in the study geographic area, self-reporting represents the only way to collect information about vaccine uptake when we exclude influenza and pneumococcal vaccinations. Furthermore, since the coverage rate revealed for all considered vaccines was suboptimal, an eventual overestimation would not determine a risk of misinterpretation of the study conclusions.

## 5. Conclusions

In conclusion, our results have shown that despite specific recommendations [13], vaccination coverage rates are far below international targets for frail subjects [3]. Our own multivariable analysis does not provide evidence for specific, immediately modifiable factors that might increase vaccination uptake or willingness to receive vaccination. However, evidence from the literature suggests reducing missed opportunities for vaccination could be a useful strategy to increase vaccination coverage in frail patients who, during the routine checks performed by GPs and specialists, should be assessed for their vaccination status and, at the same time, receive recommendation and actively offer vaccinations. To achieve this goal, it would be necessary to improve vaccine confidence among physicians and to expand vaccine service delivery to promote synergy between treatment and preventive services. As highlighted in a previous study, performed with the aim of evaluating the impact of the implementation of an on-site vaccination-dedicated clinic on the vaccination coverage rates of HCWs in a teaching hospital [47], there was an extraordinary increase in the uptake of all the recommended vaccines among HCWs. Therefore, it is possible to hypothesise that the same result could also be achieved in frail patients who could receive the vaccinations in the same healthcare facility they access for specialist visits. This is what already occurs for the influenza and pneumococcal vaccination, which are administered to frail patients directly by the GP in his/her office.

## Figures and Tables

**Table 1 vaccines-10-01311-t001:** Demographic and baseline characteristics of the study population and their association with recommended VPD vaccination uptake and willingness to get vaccinated.

Characteristic			Recommended VPD Vaccination Uptake		Total		Willingness Towards Recommended VPD Vaccination
	Tot N. (315)	%	Yes N. (66)	%	N (427)	%	Yes N. (%)
*Socio-demographic profile and behavioral risk factor*							
**Gender**							
Male	129	40.9	29	22.5	171	40.1	129 (75.4)
Female	186	59.1	37	19.9	256	59.9	181 (70.7)
			χ^2^= 0.308, 1 df, *p* = 0.579				χ^2^= 1.16, 1 df, *p* = 0.282
**Age, years Mean ± SD**	68.6 ± 12.9		67.8 ± 16.1		67.6 ± 12.8		67.3 ± 13
			*t*-test = 0.579, 313 df, *p* = 0.568				*t*-test = −0.835, 425 df, *p* = 0.403
20–55	38	12.1	11	28.9	62	14.5	48 (77.4)
56–65	59	18.7	8	13.6	91	21.3	62 (68.1)
66–75	129	40.9	25	19.4	164	38.4	125 (76.2)
>75	89	28.3	22	24.7	110	25.8	75 (68.2)
			χ^2^= 4.36, 3 df, *p* = 0.224				χ^2^= 3.80, 3 df, *p* = 0.284
**Marital status ^a,b^**							
Married	241	76.7	45	18.7	326	77.1	239 (73.3)
Other	73	20.3	21	27.7	97	22.9	68 (70.1)
			χ^2^= 3.43, 1 df, *p* = 0.064				χ^2^= 0.38, 1 df, *p* = 0.534
**Additional persons in the household ^a^**							
None	64	20.3	18	28.1	75	18.3	51 (68)
1	160	50.8	25	15.6	206	50.2	156 (75.7)
≥2	84	28.9	22	26.2	129	31.5	95 (73.6)
			χ^2^= 6.08, 1 df, *p* = 0.048				χ^2^= 1.70, 2 df, *p* = 0.429
**Education level ^a^**							
Primary and Secondary school	115	37.5	19	16.5	161	39	112 (69.6)
High school	124	40.4	24	19.4	158	38.3	120 (75.9)
University graduate	68	22.1	21	30.9	94	22.7	68 (72.3)
			χ^2^= 5.62, 2 df, *p* = 0.060				χ^2^= 1.64, 2 df, *p* = 0.440
**Working activity**							
Retired	249	79.1	52	20.9	319	74.7	423 (76.2)
Employed	66	20.9	14	21.2	108	25.3	67 (62)
			χ^2^= 0.003, 1 df, *p* = 0.953				χ^2^= 8.11, 1 df, *p* = 0.004
**Smoking status**							
Never/past smoker	288	91.4	60	20.8	388	91.5	286 (73.7)
Current smoker	27	8.6	6	22.2	36	8.5	23 (63.9)
			χ^2^= 0.02, 1 df, *p* = 0.865				χ^2^= 1.61, 1 df, *p* = 0.205
*Anamnestic characteristics*							
**Age—adjusted Charlson comorbidity index (CCIa) Mean ± SD**	4.7 ± 1.9		4.7 ± 2		4.6 ± 1.9		4.7 ± 1.9
			*t*-test = −0.29, 313 df, *p* = 0.767				*t*-test = −0.30, 425 df, *p* = 0.763
**Number of chronic health conditions**							
<3	200	63.5	43	21.5	288	67.5	201 (69.8)
≥3	115	36.5	23	20	139	32.5	109 (78.4)
			χ^2^= 0.09, 1 df, *p* = 0.753				χ^2^= 3.50, 1 df, *p* = 0.061
**Diabetes**							
No	237	75.2	52	21.9	325	76.1	234 (72)
Yes	78	24.8	14	17.9	102	23.9	76 (74.5)
			χ^2^= 0.56, 1 df, *p* = 0.452				χ^2^ = 0.25, 1 df, *p* = 0.620
**Cardiovascular disease**							
No	70	22.2	14	20	107	25.1	72 (67.3)
Yes	245	77.8	52	21.2	320	74.9	238 (74.4)
			χ^2^= 0.04, 1 df, *p* = 0.824				χ^2^= 0.02, 1 df, *p* = 0.155
**Dysmetabolic disease**							
No	182	57.8	37	20.3	258	60.4	180 (69.8)
Yes	133	42.2	29	21.8	169	39.6	130 (76.9)
			χ^2^ = 0.10, 1 df, *p* = 0.751				χ^2^ = 2.63, 1 df, *p* = 0.105
**Kidney disease**							
No	288	91.4	61	21.2	392	91.8	280 (71.4)
Yes	27	8.6	5	18.5	35	8.2	30 (85.7)
			χ^2^ = 0.10, 1 df, *p* = 0.745				*Fisher exact* =3.296,1 df, *p* = 0.069
**Respiratory disease**							
No	275	87.3	53	19.3	374	87.6	270 (72.2)
Yes	40	12.7	13	32.5	53	12.4	40 (75.5)
			χ^2^ = 3.68, 1 df, *p* = 0.055				χ^2^ = 0.25, 1 df, *p* = 0.616
**Cancer**							
No	231	73.3	56	24.2	307	71.9	222 (72.3)
Yes	84	26.7	10	11.90	120	28.1	88 (73.3)
			χ^2^ = 5.66, 1 df, *p* = 0.017				χ^2^ = 0.04, 1 df, *p* = 0.832
**Autoimmune disease**							
No	261	82.9	52	19.9	353	82.7	256 (72.5)
Yes	54	17.1	14	25.9	74	17.3	54 (72)
			χ^2^ = 0.97, 1 df, *p* = 0.324				χ^2^= 0.01, 1 df, *p* = 0.937
**Neurological disease**							
No	299	94.9	65	21.7	406	95.1	296 (72.9)
Yes	16	5.1	1	6.3	21	4.9	14 (66.7)
			χ^2^ = 2.19, 1 df, *p* = 0.138				χ^2^ = 0.39, 1 df, *p* = 0.532
**Gastrointestinal disease**							
No	295	93.6	60	20.3	396	92.7	287 (72.5)
Yes	20	6.4	6	30	31	7.3	23 (74.2)
			χ^2^ = 1.05, 1 df, *p* = 0.304				χ^2^ = 0.04, 1 df, *p* = 0.836
**GP medical visits in the previous year**							
None/Telephone contact	91	28.9	21	23.1	119	27.9	78 (65.5)
≤4	128	40.6	28	21.9	187	43.8	140 (74.9)
≥5	96	30.5	17	17.7	121	28.3	92 (76)
			χ^2^ = 0.92, 2 df, *p* = 0.630				χ^2^ = 4.17, 2 df, *p* = 0.124
*Perception of the risk of contracting VPD and attitudes towards vaccines*							
**Perceived risk of contracting VPD ^a,b^**							
Low (1–4)	98	35.9	20	20.4	141	37.6	98 (69.5)
Moderate (5–7)	87	31.9	21	24.1	122	32.5	98 (80.3)
High (8–10)	88	32.2	17	19.3	112	29.9	78 (69.6)
			χ^2^ = 0.67, 2 df, *p* = 0.715				χ^2^= 4.84, 2 df, *p* = 0.089
**Having received vaccination information**							
No	NA	NA	NA	NA	112	26.2	65 (58)
Yes	NA	NA	NA	NA	315	73.8	245 (77.8)
							χ^2^= 16.18, 2 df, *p* < 0.001
**Advise to carry out vaccinations versus VPD ^a^**							
General Practitioner	236	75.2	43	18.2	237	75.2	186 (78.5)
Specialist	46	14.6	14	30.4	46	14.6	35 (76.1)
Relatives/own decision	32	10.2	9	28.1	32	10.2	24 (75)
			χ^2^ = 4.54, 2 df, *p* = 0.103				χ^2^= 0.28, 2 df, *p* = 0.866
**Fear of vaccine side effects ^b^**							
Low (1–4)	176	55.9	39	22.2	242	56.7	185 (76.4)
Moderate (5–7)	72	22.9	19	26.4	96	22.5	74 (77.1)
High (8–10)	67	21.2	8	11.9	89	20.8	51 (57.3)
			χ^2^ = 4.72, 2 df, *p* = 0.094				χ^2^= 13.23, 2 df, *p* = 0.001
**Perceived risk of contracting SARS-CoV-2 infection ^a,b^**							
Low (1–4)	92	29.7	20	21.7	124	29.4	92 (74.2)
Moderate (5–7)	92	29.7	16	17.4	122	29	90 (73.8)
High (8–10)	126	40.6	30	23.8	175	41.6	124 (70.9)
			χ^2^ = 1.32, 2 df, *p* = 0.516				χ^2^= 0.51, 2 df, *p* = 0.775
**Advise to carry out the COVID-19 vaccination ^a^**							
General Practitioner	52	17.1	11	21.2	68	16.2	51 (75)
Specialist	169	55.4	35	20.7	246	58.4	178 (72.4)
Relatives/own decision	84	27.5	20	23.8	107	25.4	78 (74.9)
			χ^2^ = 0.32, 2 df, *p* = 0.849				χ^2^= 0.19, 2 df, *p* = 0.910
*Vaccination status*							
**Influenza vaccine (at least once in the previous three years)**							
No/I do not remember	NA	NA	NA	NA	126	29.5	67 (53.2)
Yes	NA	NA	NA	NA	301	70.5	243 (80.7)
							χ^2^= 33.91, 1 df, *p* < 0.001
**PCV13 (At least once in the previous five years)**							
No/I do not remember	NA	NA	NA	NA	330	78.9	231 (68.1)
Yes	NA	NA	NA	NA	88	21.1	79 (89.8)
							χ^2^= 16.43, 1 df, *p* < 0.001

SD: standard deviation. VPD: vaccine preventable diseases. NA: not applicable. ^a^ The numbers that do not add to the total subjects are due to missing data for variables. ^b^ A 10-point Likert-type scale ranging from 1 for “low” to 10 for “high”. Text in bold indicates the outcome variable.

**Table 2 vaccines-10-01311-t002:** Self-reported vaccination coverage prevalence of the study population.

Vaccinations	N.	%	95% CI
**Influenza vaccine (At least once in the previous three years) (427) ^a^**			
No/I don’t remember	126	29.5	0.25–0.34
Yes	301	70.5	0.66–0.75
**PCV13 (At least once in the previous five years) (418) ^a^**			
No/I don’t remember	330	78.9	0.75–0.83
Yes	88	21.1	0.17–0.25
**TDaP (427) ^a^**			
No/I don’t remember	413	96.7	0.95–0.98
Yes	14	3.3	0.02–0.05
**HZV (408) ^a^**			
No/I don’t remember	406	99.5	0.98–0.99
Yes	2	0.5	0.01–0.02
**HBV (60) ^a^**			
No/I don’t remember	55	91.7	0.82–0.97
Yes	5	8.3	0.03–0.18
**HiB (130) ^a^**			
No/I don’t remember	128	98.4	0.95–0.99
Yes	2	1.6	0.01–0.05
**Meningococcus (158) ^a^**			
No/I don’t remember	154	97.5	0.94–0.99
Yes	4	2.5	0.01–0.06
**MMRV (132) ^a^**			
No/I don’t remember	123	93.2	0.87–0.97
Yes	9	6.8	0.03–0.13

^a^ Eligible subjects. Text in bold indicates the outcome variable.

**Table 3 vaccines-10-01311-t003:** Multiple logistic regression analysis results to estimate the associations of recommended VPD vaccination uptake in frail subjects with several variables.

Variable	OR	SE	95% CI	*p* Value
**Model 1. Outcome: Recommended VPD vaccination uptake** *Log-likelihood = −136.87*, *Chi-square = 34.15*, *p = 0.0001*, *N of obs = 299*				
**Age, years (continuous)**	1.02	0.01	0.99–1.04	0.344
**Willingness towards recommended VPD vaccination**				
No	1.00 *			
Yes	3.55	1.70	1.39–9.07	*0.008*
**Additional persons in the household**				
None	1.00 *			
1	0.52	0.17	0.28–0.97	*0.041*
≥2			*Backward elimination*	
**Education level**				
Primary and Secondary school	1.00 *			
High school			*Backward elimination*	
University graduate	2.03	0.70	1.03–3.97	*0.040*
**Neurological disease**				
No	1.00 *			
Yes	0.31	0.34	0.04–2.56	0.278
**Oncological disease**				
No	1.00 *			
Yes	0.39	0.16	0.18–0.87	*0.021*
**Advise to carry out vaccinations versus VPD**				
GP	1.00 *			
Specialist	2.21	0.92	0.97–5.02	0.059
Relatives/own decision	1.66	0.81	0.64–4.32	0.301
**Respiratory disease**				
No	1.00 *			
Yes	1.94	0.78	0.88–4.28	0.100

SD: standard deviation; GP: general practitioner; VPD: vaccine preventable diseases. * Reference category. The following variables were removed from the model by the backward elimination procedure: Gender; Marital status; Total chronic health conditions; GP medical visits in the previous year. Text in bold indicates the outcome variable, whereas the text in italics are the results of the logistic model.

**Table 4 vaccines-10-01311-t004:** Multiple logistic regression analysis results to estimate the associations between willingness to receive recommended VPD vaccination in frail subjects and a variety of variables.

Variable	OR	SE	95% CI	*p* Value
**Model 2. Outcome: Willingness towards recommended VPD vaccination** *Log-likelihood = −178.08, Chi-square = 80.78, p = 0.000, N of obs = 375*				
**Age, years (continuous)**	0.93	0.01	0.91–0.97	*<0.001*
**Working activity**				
No	1.00 *			
Yes	0.40	0.13	0.20–0.78	*0.007*
**Smoking status**				
Never/ex-smoker	1.00 *			
Current smoker	0.49	0.21	0.21–1.15	0.101
**Cardiovascular disease**				
No	1.00 *			
Yes	1.54	0.50	0.82–2.89	0.182
**Kidney disease**				
No	1.00 *			
Si	3.26	1.94	1.01–10.49	*0.048*
**GP medical visits in the previous year**				
None/Telephone contact	1.00 *			
≤4	1.30	0.35	0.76–2.22	0.334
≥5	*Backward elimination*			
**Perceived risk of contracting VPD**				
Low (1–4)	1.00 *			
Moderate (5–7)	1.84	0.55	1.02–3.32	*0.042*
High (8–10)	*Backward elimination*			
**Having received vaccination information**				
No	1.00 *			
Yes	1.34	0.44	0.70–2.56	0.376
**Fear of vaccine side effects**				
Low (1–4)	1.00 *			
Moderate (5–7)	*Backward elimination*			
High (8–10)	0.38	0.11	0.21–0.68	*0.001*
**Influenza vaccine (At least once in the previous three years)**				
No	1.00 *			
Yes	3.38	1.11	1.77–6.45	*<0.001*
**Pneumo vaccine (At least once in the previous five years)**				
No	1.00 *			
Yes	3.12	1.43	1.28–7.65	*0.012*

SD: standard deviation; GP: general practitioner; VPD: vaccine preventable diseases. * Reference category. The following variable was removed from the model by the backward elimination procedure: Gender; Total chronic health conditions; Dysmetabolic disease. Text in bold indicates the outcome variable, whereas the text in italics are the results of the logistic model.

## Data Availability

The data presented in the study are available on request from the corresponding author.

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
