# Peer review of "Vaccinations Status against Vaccine-Preventable Diseases and Willingness to Be Vaccinated in an Italian Sample of Frail Subjects"

_vaccines, 2022, doi:10.3390/vaccines10081311_

Round 1
Reviewer 1 Report
This study investigates the prevalence and predictors of the willingness to receive recommended vaccinations in Italy. The paper sounds interesting, quite organized and comprehensive. Vaccination against COVID-19 has raised many concerns in public opinion. I think that it is a very relevant topic that must be addressed. The design of the study is good. I only have some suggestions:
-“immune-compromising medical conditions or at increased risk of disease due to immunosenescence” (line 50): add a consideration for people taking immunosuppressive agents. The same applies for line 71.
- This study examines vaccination coverage rates in frail subjects; however, some special populations should be discussed. What about patients affected by autoimmune disease? These populations should be mentioned as well as “older adults aged >60, immunocompromised or subjects affected by chronic conditions” (line 69). Consider the role of vaccination in special populations and autoimmune neuromuscular disease, such as myasthenia gravis. Read and cite recent papers facing this relevant topic (https://doi.org/10.3390/neurolint14020033). Deaths in unvaccinated myasthenic patients are significantly higher than in general population.
-“vaccine information” is an essential topic to be assessed. I sincerely thank the authors for including this analysis in this interesting paper. The role of GPs in general medical conditions and the one of specialists for rare diseases are both crucial. Many concerns have arisen during pandemic in patients affected by immunological condition regarding vaccination (i.e. Guillan-barré syndormes, myasthenia gravis, multiple sclerosis) and the lack of data on this topic contributed to fuel the fear of vaccination, especially from GPs who were not aware of autoimmune conditions and their management. The case of myasthenia gravis is emblematic: in this condition, the lack of vaccination indirectly killed many patients who decided not to receive vaccination and faced COVID-19 without any protection, with consequent ICU admission and respiratory failure. Conversely, vaccinated myasthenic patients with subsequent COVID-19 presented lower rates of ICU admission and no deaths. Read and cite a recent paper facing this relevant topic (Lupica et al 2022, Neurol int, suggested before). I suggest stressing this concept and encouraging studies deepening the safety of vaccination in frail patients affected by autoimmune disease.
-the design of the study is in line with the aims. The results have been discussed in full.
-there are no relevant grammar mistakes.
Author Response
Reviewer 1:
This study investigates the prevalence and predictors of the willingness to receive recommended vaccinations in Italy. The paper sounds interesting, quite organized and comprehensive. Vaccination against COVID-19 has raised many concerns in public opinion. I think that it is a very relevant topic that must be addressed. The design of the study is good. I only have some suggestions:
-“immune-compromising medical conditions or at increased risk of disease due to immunosenescence” (line 50): add a consideration for people taking immunosuppressive agents. The same applies for line 71.
We modified the introduction section (page 2, lines 54-56) and the “Study design and population” paragraph (page 3, lines 85,86) according to the suggestion.
- This study examines vaccination coverage rates in frail subjects; however, some special populations should be discussed. What about patients affected by autoimmune disease? These populations should be mentioned as well as “older adults aged >60, immunocompromised or subjects affected by chronic conditions” (line 69). Consider the role of vaccination in special populations and autoimmune neuromuscular disease, such as myasthenia gravis. Read and cite recent papers facing this relevant topic (https://doi.org/10.3390/neurolint14020033). Deaths in unvaccinated myasthenic patients are significantly higher than in general population.
In response to the point, we agree with the reviewer comment about the role of vaccination in patients affected by autoimmune disease and we accordingly added a consideration in the introduction section (page 2, lines 59-64).
- “vaccine information” is an essential topic to be assessed. I sincerely thank the authors for including this analysis in this interesting paper. The role of GPs in general medical conditions and the one of specialists for rare diseases are both crucial. Many concerns have arisen during pandemic in patients affected by immunological condition regarding vaccination (i.e. Guillan-barré syndormes, myasthenia gravis, multiple sclerosis) and the lack of data on this topic contributed to fuel the fear of vaccination, especially from GPs who were not aware of autoimmune conditions and their management. The case of myasthenia gravis is emblematic: in this condition, the lack of vaccination indirectly killed many patients who decided not to receive vaccination and faced COVID-19 without any protection, with consequent ICU admission and respiratory failure. Conversely, vaccinated myasthenic patients with subsequent COVID-19 presented lower rates of ICU admission and no deaths. Read and cite a recent paper facing this relevant topic (Lupica et al 2022, Neurol int, suggested before). I suggest stressing this concept and encouraging studies deepening the safety of vaccination in frail patients affected by autoimmune disease.
As suggested we added a comment in the discussion section (page 6, lines 313-321)
-the design of the study is in line with the aims. The results have been discussed in full.
-there are no relevant grammar mistakes.
Reviewer 2 Report
This study might have collected useful data, however the report is poorly focused, the data analysis is poorly planned, and an inappropriate selection of data are presented. I advise that this manuscript is not suitable for publication.
The objectives of the study are not immediately clear. The title is not consistent with the statements in the abstract and the introduction: "to investigate the vaccination status of frail adults during the SARS-CoV-2 pandemic and, for the subjects eligible for at least one vaccine, to explore the willingness to be vaccinated." Initially, one is led to believe that this is about COVID-19 vaccination, but later, in the statistical analysis description, the authors clarify that this is about vaccination against influenza, pneumococcus, meningococcus, pertussis, tetanus, diphtheria, Hepatitis B, Haemophilus influenzae type b, Herpes Zoster, measles, mumps, rubella, and varicella.
With respect to the first objective (regarding vaccination status), there is one paragraph (L143-149) in the Results section that states the self-reported prevalences of vaccination against the above-listed diseases (other than COVID-19). However, there is no analysis of potential predictors of vaccination status (demographics, comorbidities, health care utilization and attitudes). This is a serious omission, as this would have been potentially the most useful part of the study.
With respect to the second objective (regarding "willingness to be vaccinated"), this is not very interesting or useful. "Willingness to be vaccinated" is not clearly defined: to what vaccinations does it apply, some of them, or all of them, and under what circumstances? On its face, this seems to have no validity as an outcome; perhaps it might, if the subjects were asked such a question in a clinic that would immediately provide a specific vaccination if they stated willingness to accept, however, this was not the case, as the subjects were attendees of a clinic for COVID-19 vaccination. I understand that self-reported vaccination status is not ideal as a measured outcome if not objectively verified from health care records, but it seems better that stated willingness, which is neither verifiable nor credible. The analysis of predictors of stated willingness, which takes up most of this report, seems irrelevant.
Author Response
This study might have collected useful data, however the report is poorly focused, the data analysis is poorly planned, and an inappropriate selection of data are presented. I advise that this manuscript is not suitable for publication.
The objectives of the study are not immediately clear. The title is not consistent with the statements in the abstract and the introduction: "to investigate the vaccination status of frail adults during the SARS-CoV-2 pandemic and, for the subjects eligible for at least one vaccine, to explore the willingness to be vaccinated." Initially, one is led to believe that this is about COVID-19 vaccination, but later, in the statistical analysis description, the authors clarify that this is about vaccination against influenza, pneumococcus, meningococcus, pertussis, tetanus, diphtheria, Hepatitis B, Haemophilus influenzae type b, Herpes Zoster, measles, mumps, rubella, and varicella.
In response to the point, we have modified the manuscript title focusing the attention on the vaccination’s status against vaccine-preventable diseases (VPD) among frail subjects and, we have also clarified it in the aim of the study (abstract section: lines 17-20; Introduction section, line 71-73). Relating to the methods, it is usefulness to draw the attention on the organization of the following sections: in the first paragraph (2.1 Study design and population, page 2, lines 75-87) we have described how the study population was recruited among patients attending in a COVID-19 vaccination clinic waiting room; in the second paragraph (2.2 Survey instrument, page, 2-3, lines 88-108) we have described the questionnaire used to collect the data focusing attention to the vaccination status against VPD recommended vaccination, in line with the NPVP, among those indicated for the clinical conditions of the subjects; finally, in the third paragraph (2.3 Statistical analysis, page 3, lines 111-136) statistical analysis procedures were explained.
With respect to the first objective (regarding vaccination status), there is one paragraph (L143-149) in the Results section that states the self-reported prevalences of vaccination against the above-listed diseases (other than COVID-19). However, there is no analysis of potential predictors of vaccination status (demographics, comorbidities, health care utilization and attitudes). This is a serious omission, as this would have been potentially the most useful part of the study.
In response to the point, we agree with the reviewer comment that it would be interesting to add information regarding potential predictors of vaccination status, therefore we performed univariate and multivariate logistic regression analyses to determine the independent association of explanatory variables with recommended (VPD) vaccinations uptake. Accordingly we modified the methods section, statistical analysis paragraph (page 3, lines 118-136), the results section (page 10, lines 194-196 and lines 205-207), the discussion section (page 15, lines 301-305 ) and Tables (pages 5-9 and 12-13).
With respect to the second objective (regarding "willingness to be vaccinated"), this is not very interesting or useful. "Willingness to be vaccinated" is not clearly defined: to what vaccinations does it apply, some of them, or all of them, and under what circumstances? On its face, this seems to have no validity as an outcome; perhaps it might, if the subjects were asked such a question in a clinic that would immediately provide a specific vaccination if they stated willingness to accept, however, this was not the case, as the subjects were attendees of a clinic for COVID-19 vaccination. I understand that self-reported vaccination status is not ideal as a measured outcome if not objectively verified from health care records, but it seems better that stated willingness, which is neither verifiable nor credible. The analysis of predictors of stated willingness, which takes up most of this report, seems irrelevant.
In response to the point, we specified in method section, statistical analysis paragraph (page 3, lines 114-115) that the willingness to be vaccinated is applied to the recommended vaccination in line with the Italian National Vaccination Prevention Plan among those indicated for their clinical conditions.
Reviewer 3 Report
This is an interesting manuscript with some important and novel contributions to the literature. The authors explore the willingness to obtain vaccination in Italy.
While the paper has some merits, it needs some significant revisions before it can be considered for publication in VACCINES. Authors need to further review the extant COVID-19 literature. A relevant study recommended is:
Ramkissoon, H. (2022). COVID-19 Adaptive Interventions: Implications for Wellbeing and Quality-of-Life. Frontiers in Psychology, 13.
The methodology is sound. Authors present a robust study. Findings however need to be further tied to the extant literature (relevant studies as recommended above). Authors also need to think further on the impact of their study and elaborate on their discussion. Overall, this is an important study and can attract good readership if carefully revised. I recommend major revisions, good luck. Thank you.
Author Response
This is an interesting manuscript with some important and novel contributions to the literature. The authors explore the willingness to obtain vaccination in Italy.
While the paper has some merits, it needs some significant revisions before it can be considered for publication in VACCINES. Authors need to further review the extant COVID-19 literature. A relevant study recommended is:
Ramkissoon, H. (2022). COVID-19 Adaptive Interventions: Implications for Wellbeing and Quality-of-Life. Frontiers in Psychology, 13.
The methodology is sound. Authors present a robust study. Findings however need to be further tied to the extant literature (relevant studies as recommended above). Authors also need to think further on the impact of their study and elaborate on their discussion. Overall, this is an important study and can attract good readership if carefully revised. I recommend major revisions, good luck. Thank you.
As suggested, we added a comment in the discussion section (page 16, lines 333-340).
We checked the manuscript for language usage, spelling and grammar.
Round 2
Reviewer 2 Report
The data analysis has been improved, but the manuscript still requires major editing to present the results adequately and clearly to readers.
Abstract
L25-28: replace the sentence beginning with "Recommended VPD vaccination" with two new sentences: "In a multivariable logistic regression model, higher odds of recommended VPD vaccination uptake (defined as having two or more of the recommended vaccinations) were associated with university education (Odds Ratio=2.0, 95% Confidence Interval: 1.03 to 3.8), but history of oncological disease was predictive of lower odds of vaccination uptake (OR=0.40, 95% CI: 0.18 to 0.87). In another multivariable model, higher odds of willingness to receive vaccines were associated with kidney disease (OR=3.3, 95% CI: 1.01 to 10.5), perceived risk of VPD (OR=1.9, 95% CI: 1.02 to 3.3), previous influenza vaccination (OR=3.4, 95% CI: 1.8 to 6.5), and previous pneumococcal vaccination (OR=3.1, 95% CI: 1.3 to 7.7), but increasing age (OR=0.93 per year, 95% CI: 0.91 to 0.97), working (OR=0.40, 95% CI: 0.20 to 0.78), and fear of vaccine side effects (OR=0.38, 95% CI: 0.21 to 0.68) were predictive of lower odds of willingness to receive vaccines."
Introduction:
L70: replace "VDP" with "VPD".
Materials and methods:
L130: Replace "hausehold" with "household".
Results:
Table 2: include "willingness towards vaccination" in the list of study subject characteristics, and test the association of "willingness towards vaccination" with vaccination uptake. As a reader, I want to know if willingness (as you have measured it) has any predictive validity for vaccination uptake.
L170-176: Put the self-reported vaccination coverage prevalences into a new table, showing the N (yes, no and "don't know or no reply". Calculate the % coverage as yes/(yes+no). Show the 95% confidence intervals for the % coverage. These are important facts that readers will want to know.
L194-204: Delete the first three sentences of this paragraph, as they are unnecessary and confusing. A chi-squared test of a 2xn table does not indicate if any one category is has higher prevalence than another, they merely suggest association between the row variable and the column variable, with no measurement of the magnitude or direction of the association. Also, these are univariate associations, they could be due to confounding, so they need to be confirmed by multivariate modelling. The purpose of the univariate analysis is to identify candidate variables to be included and tested as independent variables in the subsequent multivariate models, with backwards elimination of variables. The conclusions should be based on the variables remaining in the final multivariate model. I advise replacing the deleted sentences with one sentence to the effect of, "Table 1 also shows results of the univariate analyses, testing associations between characteristics of the study subjects (as hypothesized predictors), and self-reported vaccine uptake, and stated willingness to be vaccinated (as outcomes)."
L204-214: Delete the last two sentences of this paragraph as they are difficult to understand. I advise replacing these with the following,
"In the final multivariable logistic regression model (Table 2), higher odds of VPD vaccination uptake (defined as having two or more of the recommended vaccinations) were associated with university education (Odds Ratio=2.0, 95% Confidence Interval: 1.03 to 3.8), but history of oncological disease was predictive of lower odds of vaccination uptake (OR=0.40, 95% CI: 0.18 to 0.87). In another multivariable model (Table 3), higher odds of willingness to receive vaccines were associated with kidney disease (OR=3.3, 95% CI: 1.01 to 10.5), perceived risk of VPD (OR=1.9, 95% CI: 1.02 to 3.3), previous influenza vaccination (OR=3.4, 95% CI: 1.8 to 6.5), and previous pneumococcal vaccination (OR=3.1, 95% CI: 1.3 to 7.7), but increasing age (OR=0.93 per year, 95% CI: 0.91 to 0.97), working (OR=0.40, 95% CI: 0.20 to 0.78), and fear of vaccine side effects (OR=0.38, 95% CI: 0.21 to 0.68) were predictive of lower odds of willingness to receive vaccines."
Conclusions:
L355: Replace "Reducing MOV could be a useful strategy" with "Our own multivariable analysis does not provide evidence for specific, immediately modifiable factors that might increase vaccination uptake or willingness to receive vaccination. However, evidence from the literature suggests reducing missed opportunities for vaccination could be a useful strategy".
Author Response
Response to Reviewer 2 comments
The data analysis has been improved, but the manuscript still requires major editing to present the results adequately and clearly to readers.
Point 1. Abstract
L25-28: replace the sentence beginning with "Recommended VPD vaccination" with two new sentences: "In a multivariable logistic regression model, higher odds of recommended VPD vaccination uptake (defined as having two or more of the recommended vaccinations) were associated with university education (Odds Ratio=2.0, 95% Confidence Interval: 1.03 to 3.8), but history of oncological disease was predictive of lower odds of vaccination uptake (OR=0.40, 95% CI: 0.18 to 0.87). In another multivariable model, higher odds of willingness to receive vaccines were associated with kidney disease (OR=3.3, 95% CI: 1.01 to 10.5), perceived risk of VPD (OR=1.9, 95% CI: 1.02 to 3.3), previous influenza vaccination (OR=3.4, 95% CI: 1.8 to 6.5), and previous pneumococcal vaccination (OR=3.1, 95% CI: 1.3 to 7.7), but increasing age (OR=0.93 per year, 95% CI: 0.91 to 0.97), working (OR=0.40, 95% CI: 0.20 to 0.78), and fear of vaccine side effects (OR=0.38, 95% CI: 0.21 to 0.68) were predictive of lower odds of willingness to receive vaccines."
Response 1: As suggested, we have modified the results’ description accordingly with the reviewer’ suggestion (abstract section, page 1, lines 28-39; result section page 12, lines 235-247).
Point 2. Introduction:
L70: replace "VDP" with "VPD".
Response 2: As suggested, in the introduction section (page 2, line 81), we replaced “VDP” with “VPD”.
Point 3. Materials and methods:
L130: Replace "hausehold" with "household".
Response 3: As suggested, in methods section (page 3 line 142) we replace "hausehold" with "household".
Point 4. Results:
Table 2: include "willingness towards vaccination" in the list of study subject characteristics, and test the association of "willingness towards vaccination" with vaccination uptake. As a reader, I want to know if willingness (as you have measured it) has any predictive validity for vaccination uptake.
Response 4: In response to de point, we performed a multivariate logistic regression analyses including “willingness towards vaccinations” in the list of the study subjects characteristics to determine the association of explanatory variables with recommended (VPD) vaccinations uptake. Accordingly we modified the abstract section (page 1 lines 30-31), the statistical analysis paragraph (page 3, line 141), the results section (page 12, lines 237-238) and the Table 3 (page 13).
Point 5. L170-176: Put the self-reported vaccination coverage prevalences into a new table, showing the N (yes, no and "don't know or no reply". Calculate the % coverage as yes/(yes+no). Show the 95% confidence intervals for the % coverage. These are important facts that readers will want to know.
Response 5: As suggested we have added the table reporting the prevalences of self-reported vaccination coverage of the study population. (Table 2, page 11).
Point 6. L194-204: Delete the first three sentences of this paragraph, as they are unnecessary and confusing. A chi-squared test of a 2xn table does not indicate if any one category is has higher prevalence than another, they merely suggest association between the row variable and the column variable, with no measurement of the magnitude or direction of the association. Also, these are univariate associations, they could be due to confounding, so they need to be confirmed by multivariate modelling. The purpose of the univariate analysis is to identify candidate variables to be included and tested as independent variables in the subsequent multivariate models, with backwards elimination of variables. The conclusions should be based on the variables remaining in the final multivariate model. I advise replacing the deleted sentences with one sentence to the effect of, "Table 1 also shows results of the univariate analyses, testing associations between characteristics of the study subjects (as hypothesized predictors), and self-reported vaccine uptake, and stated willingness to be vaccinated (as outcomes)."
Response 6: In line with the suggestion, we modified the text in the results section (page 12, lines 221-224).
Point 7. L204-214: Delete the last two sentences of this paragraph as they are difficult to understand. I advise replacing these with the following,
"In the final multivariable logistic regression model (Table 2), higher odds of VPD vaccination uptake (defined as having two or more of the recommended vaccinations) were associated with university education (Odds Ratio=2.0, 95% Confidence Interval: 1.03 to 3.8), but history of oncological disease was predictive of lower odds of vaccination uptake (OR=0.40, 95% CI: 0.18 to 0.87). In another multivariable model (Table 3), higher odds of willingness to receive vaccines were associated with kidney disease (OR=3.3, 95% CI: 1.01 to 10.5), perceived risk of VPD (OR=1.9, 95% CI: 1.02 to 3.3), previous influenza vaccination (OR=3.4, 95% CI: 1.8 to 6.5), and previous pneumococcal vaccination (OR=3.1, 95% CI: 1.3 to 7.7), but increasing age (OR=0.93 per year, 95% CI: 0.91 to 0.97), working (OR=0.40, 95% CI: 0.20 to 0.78), and fear of vaccine side effects (OR=0.38, 95% CI: 0.21 to 0.68) were predictive of lower odds of willingness to receive vaccines."
Response 7: In line with the suggestion, we modified the text in the results section (page 12, lines 235-247).
Point 8. Conclusions:
L355: Replace "Reducing MOV could be a useful strategy" with "Our own multivariable analysis does not provide evidence for specific, immediately modifiable factors that might increase vaccination uptake or willingness to receive vaccination. However, evidence from the literature suggests reducing missed opportunities for vaccination could be a useful strategy".
Response 8: In line with the suggestion we modified the text in the conclusions section (page17, lines 388-392).
Reviewer 3 Report
Accept, this is a much improved version.
Author Response
Comment: Accept, this is a much improved version.
Response: Thank you for your collaboration in the improvement of the manuscript.
Round 3
Reviewer 2 Report
Some minor edits are needed before publication.
Abstract:
L30-31: Replace "but history of oncological disease was predictive of lower odds of vaccination uptake (OR=0.39, 95% CI: 0.18 to 0.87)" with:
"but having another person in the household (OR=0.52, 95% CI: 0.28 to 0.97), and history of oncological disease (OR=0.39, 95% CI: 0.18 to 0.87) were predictive of lower odds of vaccination uptake."
Statistical analysis:
L140: Replace "household ducational level" with "household, educational level".
Results:
L236: Insert "and" before "university".
L237-239: Replace "and having at least one more person in the household (OR=0.52, 95% IC: 0.28 to 0.97), but history of oncological disease was predictive of lower odds of vaccination uptake (OR=0.40, 95% CI: 0.18 to 0.87)" with:
"but having another person in the household (OR=0.52, 95% CI: 0.28 to 0.97), and history of oncological disease (OR=0.39, 95% CI: 0.18 to 0.87) were predictive of lower odds of vaccination uptake."
Author Response
In response to the points, we have modified the sentences accordingly to the reviewer' comments.
My colleagues and I are sincerely indebted with the refere for the efforts in reviewing our paper.